# Powerful turbidity currents driven by dense basal layers

Charles K. Paull[1], Peter J. Talling[2,3], Katherine L. Maier[1,4], Daniel Parsons[5], Jingping Xu[6,7], David W. Caress[1], Roberto Gwiazda[1], Eve M. Lundsten[1], Krystle Anderson[1], James P. Barry[1], Mark Chaffey[1], Tom O'Reilly[1], Kurt J. Rosenberger [1D] [4], Jenny A. Gales[3,8], Brian Kieft[1], Mary McGann[9], Steve M. Simmons[5], Mike McCann[1], Esther J. Sumner[10], Michael A. Clare[3] & Matthieu J. Cartigny [1D] [2,3]

Seafloor sediment flows (turbidity currents) are among the volumetrically most important yet least documented sediment transport processes on Earth. A scarcity of direct observations means that basic characteristics, such as whether flows are entirely dilute or driven by a dense basal layer, remain equivocal. Here we present the most detailed direct observations yet from oceanic turbidity currents. These powerful events in Monterey Canyon have frontal speeds of up to 7.2 m s$^{-1}$, and carry heavy (800 kg) objects at speeds of $\geq$4 m s$^{-1}$. We infer they consist of fast and dense near-bed layers, caused by remobilization of the seafloor, overlain by dilute clouds that outrun the dense layer. Seabed remobilization probably results from disturbance and liquefaction of loose-packed canyon-floor sand. Surprisingly, not all flows correlate with major perturbations such as storms, floods or earthquakes. We therefore provide a new view of sediment transport through submarine canyons into the deep-sea.

[1] Monterey Bay Aquarium Research Institute (MBARI), 7700 Sandholdt Rd, Moss Landing, CA 95039, USA. [2] Departments of Geography and Earth Sciences, Durham University, Lower Mountjoy, South Road, Durham DH1 3LE, UK. [3] National Oceanography Centre, University of Southampton Waterfront Campus, European Way, Southampton SO14 3ZH, UK. [4] Pacific Coastal and Marine Science Center, U.S. Geological Survey, 2885 Mission Street, Santa Cruz, CA 95060, USA. [5] Energy and Environment Institute, University of Hull, Cottingham Road, Hull HU6 7RX, UK. [6] Department of Ocean Science and Engineering, Southern University of Science and Technology of China, No 1088, Xueyuan Road, Nanshan District, 518055 Shenzhen, Guangdong, China. [7] Qingdao National Laboratory for Marine Science and Technology, 266061 Qingdao, China. [8] University of Plymouth, Drake Circus, Plymouth, Devon PL4 8AA, UK. [9] Pacific Coastal and Marine Science Center, U.S. Geological Survey, 345 Middlefield Road, MS999, Menlo Park, CA 94025, USA. [10] Ocean and Earth Science, University of Southampton, University Road, Southampton SO17 1BJ, UK. Correspondence and requests for materials should be addressed to C.K.P. (email: paull@mbari.org)

Turbidity currents deposit many of the largest sediment accumulations on Earth[1–3], and sculpt the deepest canyons on our planet[4,5]. These sediment gravity flows are important because they flush globally significant volumes of sediment[6,7] and organic carbon[8,9] into the deep ocean, thereby affecting global geochemical cycling[10] and deep-seafloor ecosystems[11]. They break valuable seabed pipelines, and communications cables that carry >95% of global data[12], while their thick deposits host important oil and gas reserves[13].

Turbidity currents are challenging to measure[1,14–16] because they occur underwater, are destructive, and it is difficult to predict when and where they will occur. Successive seafloor cable breaks show that frontal velocities of oceanic turbidity currents can be up to 19 m s$^{-1}$ [17,18]. However, understanding the anatomy of these flows requires profiles of both velocity and sediment concentration, ideally at multiple locations along their path to capture how flows evolve. While millions of such profiles exist for rivers[6,19], velocity profiles from turbidity currents have been measured in only ten sites worldwide[1]. There are even fewer direct measurements of sediment concentration in turbidity currents, even at a single height above the seabed[1].

Due to the challenges of measuring turbidity currents, our understanding of their anatomy is based primarily on the interpretation of their deposits in the geologic record, laboratory experiments, and computational[20] or analytical[21] models. However, similar deposits can be formed by different flow types[1,22,23]. Laboratory experiments[24] may not capture the dynamics of more powerful oceanic turbidity currents[25] because of the uncertainties in scaling. Numerical models depend on key assumptions or boundary conditions, such as mass exchange with the bed[20] which may be uncertain.

The fundamental structure of turbidity currents has remained unresolved despite being essential input for modeling and predicting turbidity current dynamics, their impact on seafloor infrastructure, and the architecture of their deposits. In particular, it is important to determine whether turbidity currents are dilute sediment suspensions, as is the case for most rivers; or whether turbidity currents are driven by near-bed layers with high (>10%) sediment concentrations. There are fundamental differences in how dilute suspensions, and flows with dense near-bed layers (i.e., debris flows or granular flows) behave, and what their deposits look like[22,23,25–27]. For example, near-bed sediment concentrations strongly affect the basic mechanism(s) that keep sediment aloft, basal friction coefficients, and rates of bed erosion; all of which determine driving forces, flow velocity, runout, and impact forces on seabed structures.

A second key question about turbidity currents is how they are triggered. Previous studies have mostly inferred that turbidity currents need a major external trigger, such as an earthquake, storm, or river flood[1,28], although flows may be delayed for hours to several days after a flood peak[12,37], or resulted from a combination of low tides and high river discharge[29]. This inference is important because it forms the basis for predicting when turbidity currents occur, and their recurrence intervals, which is important for hazard assessments[30]. Monitoring was conducted in Monterey Canyon, offshore California, because previous work demonstrates that multiple turbidity currents occur each year[16,31] and canyon morphology[32], and recent seafloor deposits[31,33,34] are already well characterized.

Here we present the first results from an ambitious monitoring study that reveals the detailed anatomy and timing of turbidity currents, and how they evolve between sites spread over a long (50 km) distance. Turbidity currents and their impact on seafloor morphology are characterized using a dense array of >50 sensors deployed for an 18-month period (Fig. 1), combined with precise mapping of seafloor change. We provide data that show turbidity currents in Monterey Canyon contain a fast and dense near-bed layers. We also document the occurrence of turbidity currents without major external triggers.

## Results

**Monitoring array.** An array of sensors that included six moorings spaced from 285 to 1850 mwd (meters water depth) covering a 50 km stretch of the canyon axis, were deployed in Monterey Canyon (Fig. 1). On each mooring, a downward-looking Acoustic Doppler Current Profiler (ADCP) (300 kHz frequency) was mounted 65 to 70 m above the seafloor, and usually equipped with a pressure sensor that recorded water depth. The ADCP-

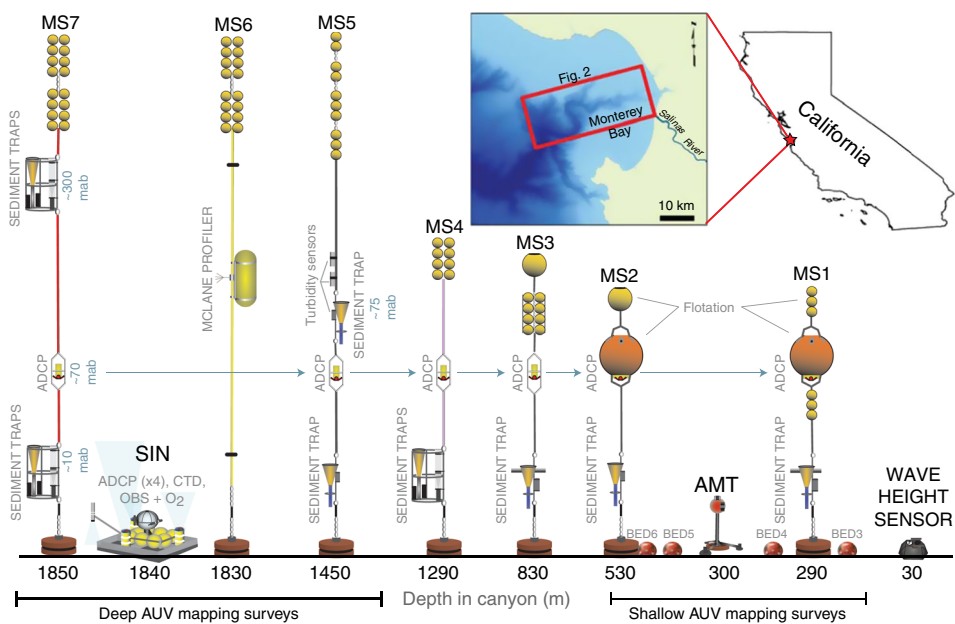

**Fig. 1** Instrument array. Schematic drawing (not to scale) showing monitoring instruments deployed within Monterey Canyon. Inset maps show location of Monterey Canyon with respect to California and area covered by map in Fig. 2. Moored ADCPs were positioned 65 to 70 m above bottom (mab)

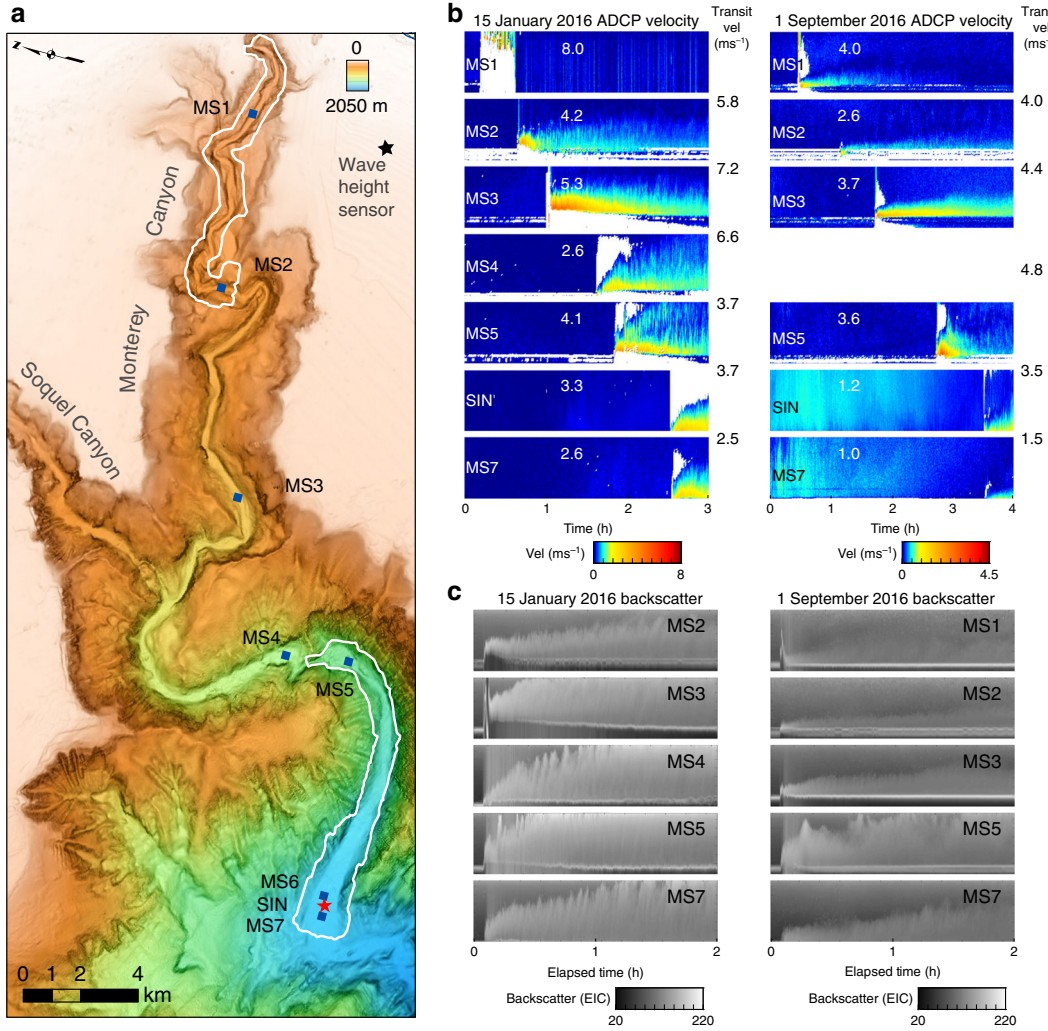

**Fig. 2** Velocity and backscatter during two through-canyon flow events. **a** Map shows mooring locations with repeat mapping areas outlined in white. **b** ADCP-measured velocities capture the arrival of the 15 January 2016 and 1 September 2016 flows as they reach successive moorings within the canyon. Unresolved velocity readings are in white. Maximum ADCP-measured velocities (white numbers) and transit velocities (black numbers) are shown for comparison. **c** ADCP-measured backscatter, beginning 5 min before the arrival of the event, shows the evolution of the flow at each mooring. Echo Intensity Counts (EIC) are proportional to decibels. The ADCPs captured a 65 to 70 m range above the seafloor when the moorings were stable and upright. Ranges to seafloor decreased when the moorings were pulled downwards during events

measured profiles of water column velocity and backscatter (a proxy for sediment concentration[35,36]) at 1 m vertical intervals, every 30 s. In addition, three upward looking ADCPs were deployed on a seabed frame (Seafloor Instrument Node (SIN)) located just upstream of the deepest mooring; these ADCPs had three different frequencies (300, 600, and 1200 kHz) in order to constrain sediment concentration profiles derived from backscatter. ADCPs are ineffective for measuring the bottom few meters of a turbidity current[33], therefore novel sensors were designed and deployed to overcome this: (i) Benthic Event Detectors (BEDs) are motion sensors encased in boulder-sized housings that are initially buried in the seafloor and then carried within flows; (ii) one of these motion sensors along with an Acoustic Monitoring Transponder (AMT), that also measures motion, was mounted on top of an 800-kg tripod frame with 1.5-m-long legs. Two segments of the canyon floor in the proximal (190 to 560 mwd) and distal parts of the sensor array (1300 to 1885 mwd) were mapped six times with exceptional vertical precision (10 cm) using Autonomous Underwater Vehicles (AUVs) to document morphological change.

**Anatomy of flow events**. The ADCP array detected 15 turbidity currents, based on sudden changes in velocity and backscatter within the water column (Fig. 2). Data from the ADCP-moorings demonstrate that turbidity currents began with a thin (<10 m), fast (typically >2 m s$^{-1}$), and dense layer for a short duration (3–10 min). Most flows lasted <30 min at the head of the array (MS1 in Fig. 1) and died out within the canyon (Fig. 3). Fourteen of the flows originated in the uppermost Monterey Canyon (<290 mwd). One event was only detected by the moorings at 1290 and 1450 mwd, which are below the intersections with Soquel Canyon, a tributary that merges with Monterey Canyon in ~995 mwd (Figs. 1, 2). The three flows that traversed the full sensor array to >1850 mwd (Fig. 3) evolved into thicker (>30 m) and longer duration (4–6 h) flows farther downstream (Fig. 2).

The front of each turbidity current caused an abrupt increase in water pressure that slowly declined (over 4 to 120 min) but did not always return to initial values. We infer that mooring cables were initially pulled abruptly downwards, before returning back to vertical as the flow waned. Irreversible pressure changes represent increases in water depth, and down-canyon transportation of

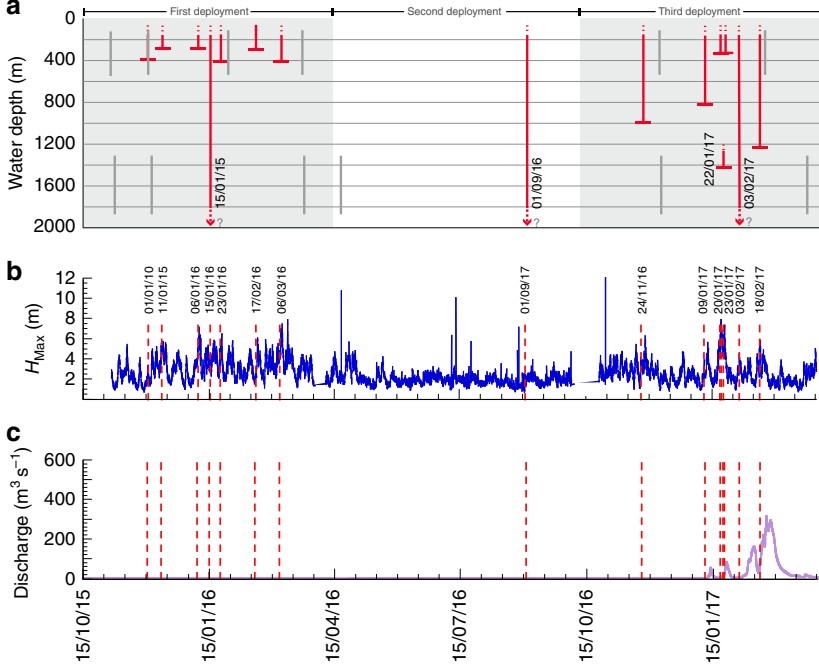

**Fig. 3** Timing of flow events compared with wave height and Salinas River discharge. **a** Occurrence of sediment density flow events and runout depth shown with red vertical lines. Three flows ran past the last mooring, and their full runout distance is unknown. The gray lines show the depth range covered and timing of repeat bathymetric surveys of the canyon floor. The three six-month-long mooring deployments are noted. **b** Blue line shows the maximum wave heights ($H_{Max}$) measured by the wave height sensor (Fig. 2, Supplementary Data 6). **c** Salinas River discharge data (Supplementary Data 7) shown in purple. Dotted red lines in **b** and **c** indicate when flow events occurred

moorings (including 450 kg anchors) (Supplementary Data 1). The most marked such event occurred on 15 January 2016 when the shallowest mooring moved 7.1 km at an average speed of 4.5 m s$^{-1}$.

**Flow front (transit) velocities between instruments**. The transit velocity of a flow front (Fig. 2) is calculated from its arrival time at sequential sensors and their deployment positions along the canyon thalweg (Supplementary Data 2). Transit velocities between successive moorings commonly exceed maximum ADCP-measured velocities (Fig. 2). Near-bed ADCP measurements are compromised during the first 1–2 min of an event by erratic mooring movements (including tilt and down-canyon displacement) and by reflections from the channel flanks in the narrow sections of the canyon. The fastest reliable ADCP-measured velocities commonly occurred several minutes after arrival of the flow front, within a high backscatter zone near the seafloor (Fig. 2).

**Repeated remobilization of objects buried within the seabed**. The turbidity currents were capable of transporting very heavy objects. The 800 kg AMT-tripod-frame moved down-canyon six-times. It moved 4.2 km on 15 January 2016 and was found on its side half-buried in the seabed. Upon re-deployment, it moved 0.9 km on 24 November 2016 (Fig. 4) and was again found on its side buried with only one foot sticking above the seafloor (Fig. 4b) entombed in an at least 2 m thick sediment layer. On both deployments, the AMT temperature sensor made measurements every 45 min, and ceased to record tidal oscillations once the AMT-frame had been transported (Fig. 4d, Supplementary Data 3). This damping of the tidal temperature signal (Fig. 4d) indicates burial of the AMT-frame beneath the seabed, although the frame might have been re-exhumed for short periods between temperature measurements. Pressure measurements show that

the 2-m tall AMT-tripod-frame moved five times after it was buried (Fig. 4d). Such movement of buried objects demonstrates remobilization of the seabed.

Transit velocities of 4.8 to 5.3 m s$^{-1}$ (Fig. 4c) during the 24 November 2016 turbidity current were calculated from the distance traveled by the event between boulder-like BEDs spaced along the canyon axis in 208 to 327 mwd. These transit velocities exceed the maximum current velocity (3.9 m s$^{-1}$) measured by the ADCP on the mooring deployed at 290 mwd during this event.

The velocity of individual BEDs is calculated from the change in pressure over time (Fig. 4c, Supplementary Data 4) converted to distance traveled based on the canyon thalweg bathymetry (Supplementary Data 5). For example, on 24 November 2016, the AMT-tripod-frame with in-water density >6 g cm$^{-3}$ moved at 4.0 m s$^{-1}$, and the nearest two BEDs with in-water densities of 2.1 g cm$^{-3}$ moved with average velocities between 2.7 and 4.0 m s$^{-1}$, despite their different size, shape and density. The BED and AMT-tripod-frame velocities were slower than the transit velocities. The synchronous movements of BEDS show flows can be active over distances of several kilometres (Fig. 4c).

**Seafloor change and deposits**. Repeat mapping surveys in the upper canyon show that substantial morphological change occurred in the areas where the heavy objects moved (Fig. 5). These maps show a 150–300 m wide swath of the active canyon channel characterized by abundant crescent-shaped bedforms with amplitudes of 1–3 m and wavelengths of 20–80 m. Changes in the position of these 1–3 m high bedforms led to widespread vertical changes of ±3 m between surveys (Fig. 5), and vertical changes exceeded 3 m in a few places. This narrow swath of bathymetric change, which is composed of poorly sorted coarse

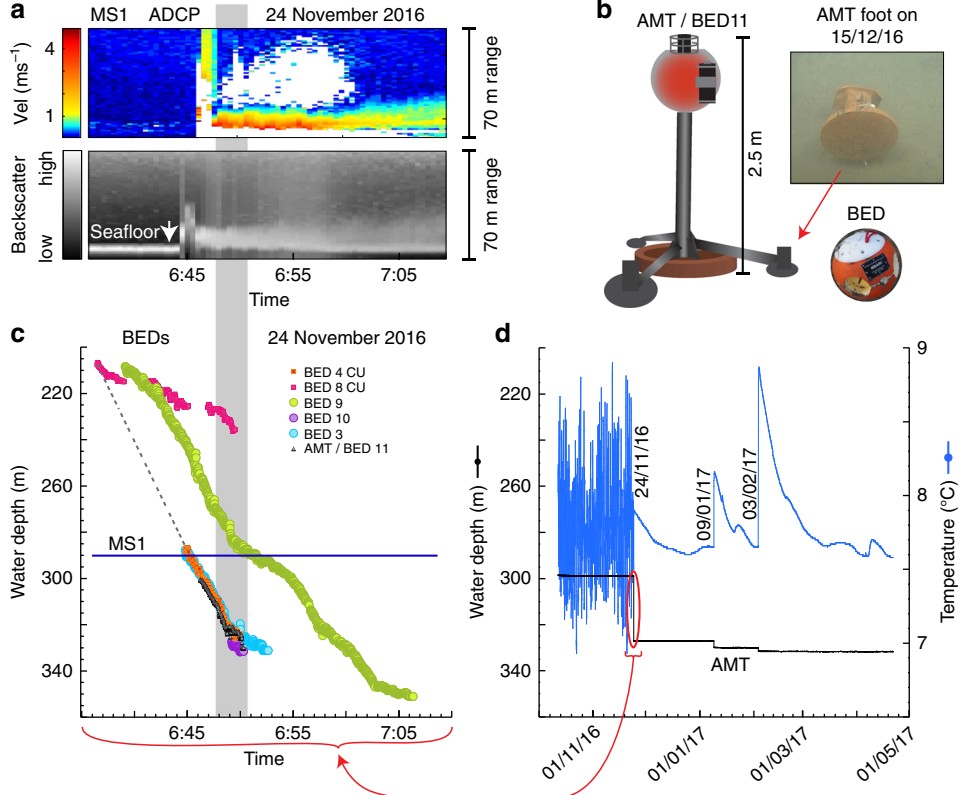

**Fig. 4** 24 November 2016 flow event records. **a** Thirty-five-minute-long record of ADCP-measured velocity and backscatter recorded on mooring MS1 during the 24 November 2016 flow event. **b** Images of a round BED and the 800-kg tripod frame with BED11 and AMT attached. Also shown is one foot of the tripod frame, exposed above the seafloor after the event. **c** Changes in depth of 6 BED instruments during the 24 November 2016 flow event for the same time period shown in **a**. Dotted line connects initial bed movements, which are used to calculate the transit velocity through this depth interval. Gray area in **a** and **c** indicates period when all six instruments were moving simultaneously. **d** Plot of depth and temperature vs. time from the AMT (sampled every 45 min) during the third deployment (Fig. 3). The range of water depth shown in **c** and **d** are equivalent. The red oval indicates the time interval shown in **c**

sand, gravel and mud-clasts[31,33,34], is where movement of the heavy objects took place (Fig. 5).

The repeat mapping survey towards the end of the sensor array (at ~1800 mwd; Fig. 5) contains similar large bedforms[32]. However, these bedforms at ~1800 mwd experienced limited (<0.5 m) crest-erosion, and lee side deposition, despite the fast (up to 3.3 m s$^{-1}$; Fig. 2) velocities of some flows. In contrast to the AMT frame in the upper canyon, the SIN frame located on the seabed at 1850 mwd was not buried, and its temperature sensors continued to record tidal oscillations.

## Discussion

The passage of turbidity currents within Monterey Canyon was measured with unprecedented precision (Fig. 2), enabling new insights into flow triggering and their internal structure. Fourteen events originated in Monterey Canyon in less than 290 mwd. The event which was first detected in 1290 mwd may have come from Soquel Canyon tributary (Fig. 2). Previous work mainly suggests that major events, such as river floods[37], earthquakes[38], or anomalously-large wave heights[39] trigger turbidity currents[28,29,40]. None of the flows documented here are linked to earthquakes (>$M_w$ 2.0) and only the last event occurred when there was any significant discharge occurring in the Salinas River (Fig. 3c), which has been engineered to enter the ocean directly at the head of Monterey Canyon under low flow conditions. Fourteen of the fifteen flows occurred in the winter months (Fig. 3).

These events typically coincide with large storm wave heights (Fig. 3b), which may have triggered seabed failure in the upper canyon. However, one of the most powerful flows (1 September 2016), which ran out at speeds of up to 5 m s$^{-1}$ (Fig. 2) through the whole sensor array (Fig. 3), occurred in a period without large wave heights, floods or earthquakes. This event suggests that turbidity currents do not always require major external triggers. Small perturbations (e.g., normal wave heights) may cause seafloor failure that produces powerful and long runout flows (Figs. 2, 3).

Flow-front transit velocities between moorings and BEDS reached up to 7.2 m s$^{-1}$, and typically exceeded the highest velocities measured by ADCPs (Figs. 2, 4). This key observation indicates that the fastest part of the flow is located within ≤2 m above the seabed, where ADCP measurements are compromised[41], or within underlying remobilized seafloor.

Although sediment concentrations were not measured directly, our observations support the existence of a dense (i.e., ≫10%[26,42] volume) remobilized layer for the following reasons. First, rafting and entombment within a dense layer of flowing sediment explains the successive exhumation, movement and burial of heavy objects (Fig. 4d). Similar successive down-canyon movements and burial were observed previously with prototypes of the BEDs[31]. It seems less likely that an entirely dilute flow, perhaps with a thin bedload layer, could transport these sometimes exceptionally heavy objects. If the 800 kg AMT-frame was entombed in a dense layer, then the thickness of that dense layer

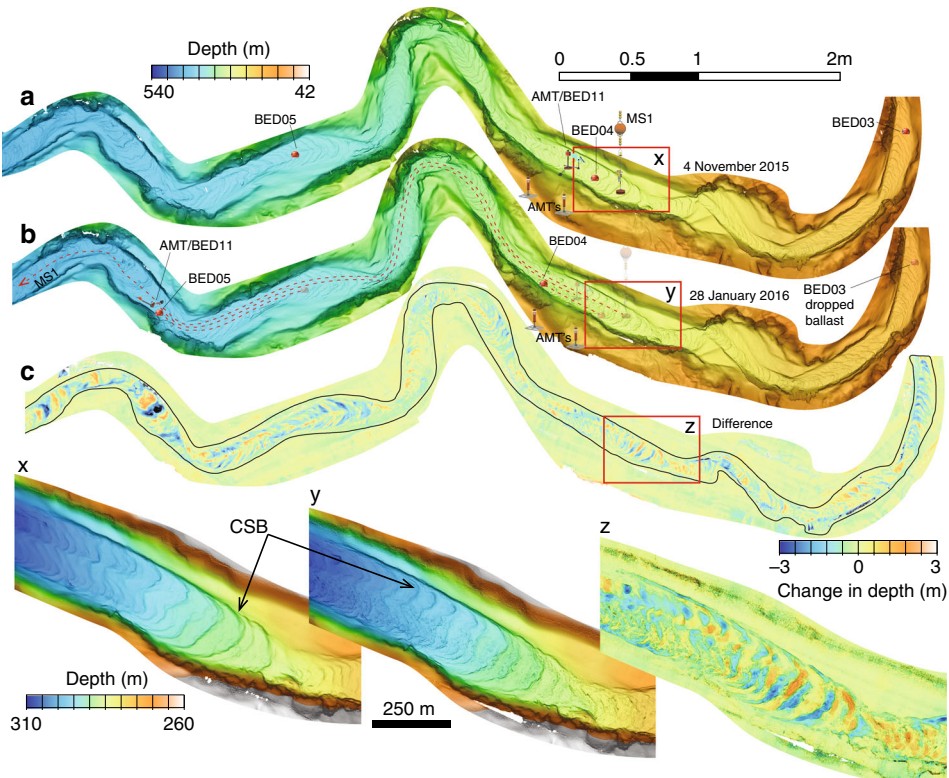

**Fig. 5** Repeat mapping shows changes in canyon floor morphology. **a** Bathymetric maps for the upper canyon collected with an Autonomous underwater vehicle between 42 and 540 m water depths on 4 November 2015 and **b** on 28 January 2016 (Fig. 1). Initial and final positions of instruments that moved on 15 January 2016 are also shown in **a** and **b**, respectively. **c** Changes in seafloor elevation between surveys **a** and **b**. x, y, and z are enlarged sections of **a**, **b**, and **c**, respectively. CSB Crescent-shaped bedforms

is at least comparable to its diameter (2 m). Second, temperature sensors on the heaviest object (800 kg AMT frame) ceased recoding tidal fluctuations, suggesting the AMT was most likely entombed within the remobilized bed, although it could have been exhumed for short periods between these measurements (Fig. 4c). Third, objects with very different size, shape and densities moved at broadly similar speeds down-canyon behind the flow front (Fig. 4b). This includes an irregularly shaped 800 kg AMT-frame, and much smaller and less dense BEDs (Fig. 4b). This is more consistent with rafting than being dragged beneath a dilute flow, where such objects with different size, shape and densities would be expected to travel at different speeds. Finally, flows that moved heavy objects are often <15 m thick, as documented by ADCP data (Fig. 4a). If these flows are entirely dilute, they are unlikely to displace entire >80 m high moorings with 450 kg anchors (Figs. 1, 2). Their motion is better explained by the anchors being rafted in a dense layer, rather than by drag on the mooring cable from a relatively-thin, dilute flow.

We lack detailed in situ seabed measurements of how dense remobilized layers originate. However, the floor of Monterey Canyon often comprises loose-packed sand that is susceptible to failure and liquefaction[22,43]. Indeed, liquefaction of canyon floor sand has been observed to be induced by vibration during coring operations (see ref. [44] supplementary video), or by divers[45]. Detailed measurements from partially water-saturated sediment below terrestrial debris flows with similar (4–15 m s$^{-1}$) speeds are also informative[46,47]. They emphasize how contractive shear displacement of loose-packed substrates, and liquefaction, have a key role in substrate remobilization, as well as reducing basal friction[46,47]. Sudden undrained loading produces high pore pressures beneath the front of these large-scale subaerial debris

flows, which erode the partly-water-saturated substrate at their front, such that the debris flow accelerates and is self-sustaining. The substrate on the floor of Monterey Canyon is fully water-saturated, and for reasonable values of sand permeability and basal shear rates, high pore pressures are likely to develop during flows[47]. We thus infer that liquefaction of loose-packed sand may have an important role in producing the fast-moving dense remobilized layer at the base of the turbidity current.

Models of submarine flows with a dense remobilized layer (Fig. 6a) must be consistent with the existence of crescent-shaped bedforms, which are ubiquitous along the floor of Monterey Canyon (Fig. 5)[32,48]. These bedforms have heights of 1 to 3 m, and wavelengths of 20 to 80 m (Fig. 5). Similar bedforms occur in many other sandy submarine canyons and channels worldwide[32,48–50]. They have been attributed to flow instabilities (termed cyclic steps) that develop within supercritical flows, which lead to hydraulic jumps and trains of up-slope migrating bedforms[30,35,49–53]. What is unknown is whether the bedforms are preserved during the dense sediment current processes observed here, which appear to remobilize the bed potentially through liquefaction.

The motion data recorded by the BEDs, as they were carried down canyon, provide important information about when bedforms are present (Figs. 4, 7). The movements of individual BEDs probably reflect conditions a short distance behind the flow front (Fig. 6). Pressure records indicate BEDs often experienced vertical oscillations with amplitudes of 1–3 m, even for the BED attached to the 800 kg AMT-frame (Fig. 7). The high density of this AMT-frame (>6 g cm$^{-3}$) suggests that it would move along the base of the flow. The amplitude and wavelengths of these vertical oscillations (Fig. 7b) are broadly similar to crescent-shaped bedforms

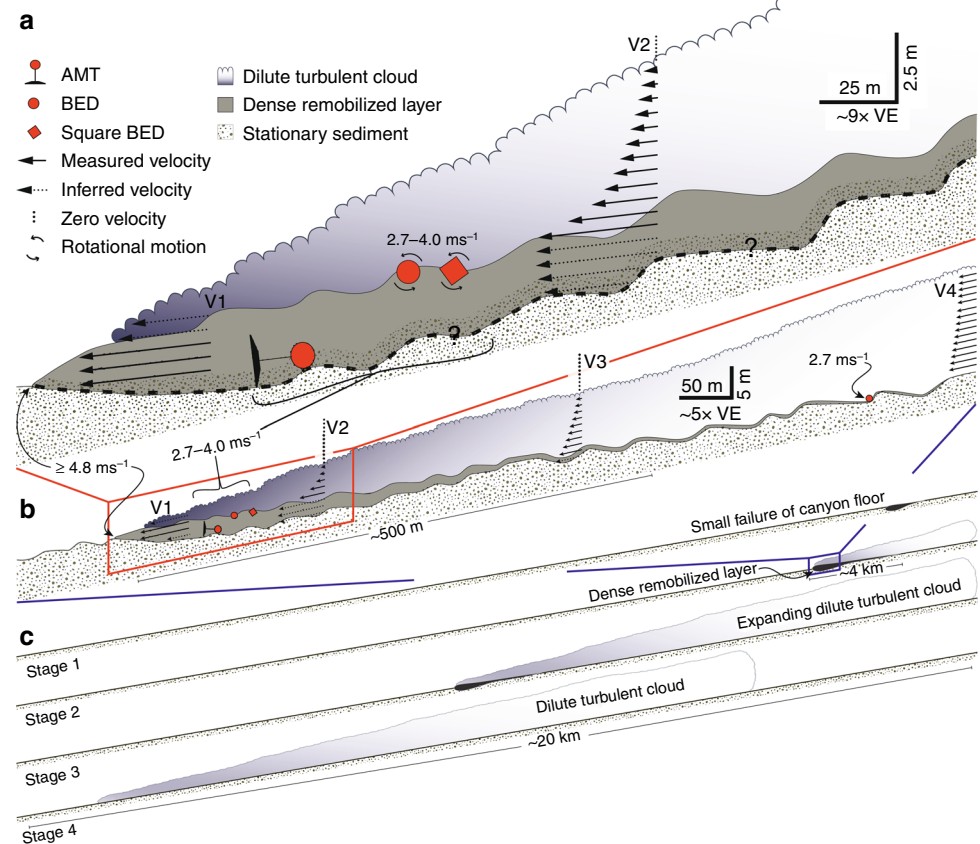

**Fig. 6** Conceptual drawings of sediment density flow events in Monterey Canyon. **a** The highest velocities (V₁) occur in a dense basal layer near the flow front. This dense basal layer forms via liquefaction or mechanical erosion of underlying loose-packed sand, and helps to generate trains of crescentic bedforms. **b** Increased turbidity in the water column is coincident with slowing of the remobilized layer. **c** The evolution of a flow as it progresses down canyon. (Stage 1) A failure in the canyon floor results in the liquefaction of the seafloor at the front of the flow, (Stage 2) it propagates down-canyon as a dense remobilized layer, (Stage 3) the fast flow progressively generates an expanding dilute turbulent cloud, (Stage 4) which continues as a dilute turbidity current

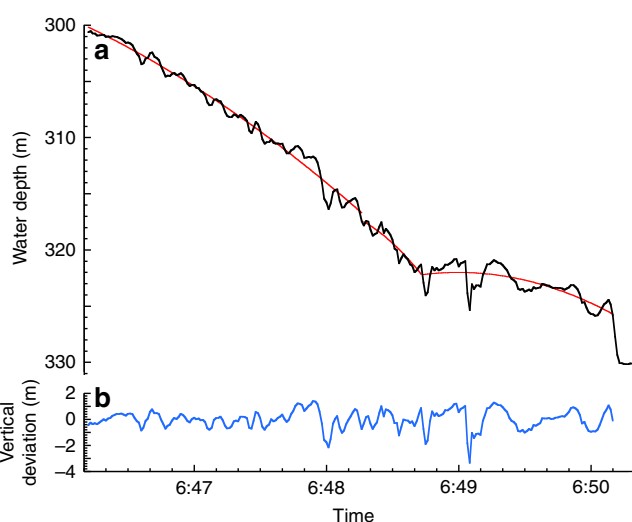

**Fig. 7** Movement of massive 800 kg frame down canyon during the 24 November 2016 event. **a** Black line shows depth (converted from pressure) vs. time from BED 11 which was attached to the 800 kg frame (AMT/BED11; Fig. 4) during the 24 November 2016 event. Red line segments are polynomial fits to three sections of these data. BED 11 traveled at an average speed of 4 m s⁻¹. **b** Deviations (blue line) of BED 11 trajectory (**a**: black line) from the fitted polynomials (**a**: red line) show vertical oscillations between 1–3 m

(Fig. 5). Thus, these oscillations suggest that the AMT-frame traveled over bedforms, which were thus not wiped-clean by frontal plowing or other erosional processes (Fig. 6).

Suitable in situ physical properties measurements (e.g., pore pressure[44,45]) to determine exact processes of erosion and bedform generation near the flow front were not collected. However, field-observations and detailed laboratory experiments show that cyclic steps and up-slope migrating bedforms can form beneath supercritical flows with very high (20–40% volume) sediment concentrations[54,55], as well as beneath dilute supercritical flows[48–52]. Previous work notes that bedform migration below dense near-bed layers can be accompanied by local bed liquefaction[54], and bedform dimensions may be controlled by properties of this dense near-bed layer[55]. We therefore propose that the frontal part of the flow liquefies (and possibly also mechanically erodes) the sandy canyon-floor, helping to sustain a dense near-bed layer below which bedforms persist and develop. Our time-lapse surveys are also too infrequent to distinguish between models in which the flow-front wipes out pre-existing bedforms, and new bedforms are created; or flow simply modifies these pre-existing bedforms (Fig. 6). Bedforms may be sculpted further by the dilute trailing body of the event, which itself may be supercritical (Fig. 6).

We conclude with a model (Fig. 6) for the evolution and anatomy of turbidity currents, based on these novel field data. Turbidity currents are initiated in the upper canyon mainly by failure within the loosely packed sand in the canyon axis or within sediment draping the flanks of the canyon (Fig. 6c). The failure and

liquefaction creates a dense fast-moving layer (dense remobilized layer) that accelerates downslope (Fig. 6b; Fig. 6c Stages 1 and 2). Erosion and liquefaction of canyon-floor sand behind the flow front produces a self-sustaining, fast and dense basal layer, which drives the overall flow-event. Migration of the crescent-shaped bedforms underneath the dense remobilized layer, as a consequence of substrate erosion on the lee side and deposition on the down-canyon stoss side, explains the ±3 m amplitude bathymetric change observed between repeat AUV surveys (Fig. 5).

Shear between the dense remobilized layer and overlying water causes mixing (Fig. 6b; Fig. 6c Stages 2 and 3) that generates an overlying dilute, turbulent sediment-suspension (Fig. 6a; $V_2$ and $V_3$). A few minutes (~2–5 min) after arrival of the flow front, the velocity of the dense remobilized layer declines (Fig. 6b; $V_4$). This is demonstrated by relaxation of the mooring cable after its initial abrupt pull down. While the initial powerful, fast, dense, remobilized layer dies out, the dilute turbulent sediment flow that it spawns can last for hours (Figs. 2 and 6b; Fig. 6c Stage 4).

Turbidity currents have previously been compared to rivers. However, our work suggests that this comparison is not always justified, as their basic structure can be fundamentally different[56,57]. Rivers are almost always entirely dilute sediment suspensions, with dense bedload layers that are only a few grains thick[58]. Rivers lack the dense remobilized layers that are several meters thick, which we document in these turbidity currents (Fig. 6). These dense basal layers can carry exceptionally heavy (800 kg) objects, at speeds of >4 m s$^{-1}$ approaching that of the flow front, for kilometres. This study also shows that powerful turbidity currents do not need major external triggers. It thus documents a new view of submarine flows that dominate sediment transfer via canyons into the deep-sea.

## Methods

**Field program**. The Coordinated Canyon Experiment deployed an array of moorings and other instruments for an 18-month period from October 2015 to April 2017 within Monterey Canyon (Fig. 1). All the moored instruments were recovered and redeployed in April 2016 and October 2016.

**Mooring data**. Six of the moorings carried oceanographic instruments and were deployed within the axis of Monterey Canyon at water depths of 285, 527, 830, 1285, 1450, and 1850 m (Fig. 1). These moorings included downward-looking Teledyne RDI Workhorse 300 kHz ADCPs mounted 65 to 70 m above the seafloor. ADCPs measured profiles of velocity and acoustic backscatter (a function of sediment concentration and grain size) through the flows at sampling rates of 7-ping ensembles every 30 s. It is important and non-trivial to determine which ADCP-measured velocities are reliable, especially near the front of flows when the moorings were experiencing rapid physical movements and when high sediment concentrations make the definition of the bottom ambiguous[41]. The maximum ADCP-measured velocity during an event was defined as the maximum down-canyon velocity measured with all four beams for at least one ping in the ensemble. During some flow events, maximum ADCP-measured velocities, determined using the above criteria, occur below the depth of maximum backscatter intensity (Fig. 2). Pressure was also recorded on three ADCPs.

**Seafloor Instrument Node**. A SIN (Fig. 1) deployed at 1840 mwd carried 300, 600, and 1200 kHz upwards-looking Teledyne RDI Workhorse ADCPs at sampling rates of 14-, 28- and 54-ping ensembles every 10 s respectively. These ADCPs were positioned ~0.5 m above the seabed and recorded on a common time base.

**Motion and displacement sensors**. Novel BED instruments were developed by MBARI to record seabed motion during down-canyon transport in flow events. BEDs contain accelerometers along three orthogonal axes, a time recorder, and a pressure sensor inside a pressure case rated to 500 mwd (Fig. 3). Rotation > 2° s$^{-1}$ triggers a recording rate of 50 Hz until the BED stops rotating. Remotely operated vehicles were used to partially bury BEDs within the canyon floor at water depths between 200 and 400 m (Fig. 1b). BEDs are usually housed in 44.5 cm diameter spheres or cubes of syntactic foam, ballasted to a density of 2.1 g cm$^{-3}$. Built-in acoustic beacons and modems allow for BEDs to be located, and data to be downloaded, even when BEDs are buried in sediment to depths of >1 m. BEDs transport velocities during flow events were calculated from the internally recorded duration of motion and a 10-s running average of pressure (i.e., depth), which were converted to distance traveled based on canyon thalweg depths[33].

A third instrument package was a Sonardyne AMT and a BED (Fig. 4) mounted on a tripod consisting of a 2-m-long post with three 1.5-m-long legs and a central bottom locomotive wheel to provide stability. The 800 kg AMT-tripod-frame was deployed in the canyon axis at ~300 mwd in October 2015 and again in October 2016. Every 45 min, the AMT measured pressure, temperature, pitch, heave, and roll.

MBARI-developed AUVs carrying Reson 400-kHz multibeam sonars conducted mapping surveys of Monterey Canyon floor and its lowermost flanks between water depths of 190–560 m and 1300–1885 m. These pre-programmed AUV missions provided exceptionally high-resolution bathymetric grids with 0.10 m vertical and 1 m horizontal resolution[59]. AUV surveys were repeated six times between October 2015 and April 2017 (Fig. 3). Maps showing changes in seafloor morphology were constructed from differences in bathymetry between successive surveys.

## Data availability
The authors declare that the data supporting the findings of this study are available within the Supplementary Data files and at https://doi.org/10.1594/IEDA/324529.

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

## Acknowledgements

The David and Lucile Packard Foundation, Natural Environment Research Council (NE/K011480/1), U.S. Geological Survey, Ocean University of China, and Qingdao National Laboratory for Marine Science and Technology provided support. Thanks to the Coordinated Canyon Experiment Team consisting of the BEDs development project group, the R/Vs *Western Flyer*, *Rachel Carson* and *Paragon* crews, ROV pilots, AUV operators, and USGS Marine Facilities. Teledyne RDI provided equipment.

## Author contributions

C.K.P., P.T., D.P. and J.X. conceived of the project and coordinated the multi-institutional resource commitments. D.W.C., M.C., T.O., B.K. and M.M. developed instruments and software necessary for this project. C.K.P., R.G., K.L.M. and K.R. coordinated the field programs. All authors discussed the results and commented on the manuscript.
