## [Peer Review File · Nature Communications]

Reviewer #1 (Remarks to the Author):

Overall Assessment

The paper reports the first results from an ambitious monitoring study conducted in Monterey Canyon (offshore California) revealing new insights on the behaviour of turbidity currents. The major claims of the paper include: a) the attempt to unravel the presence of a dense, fast, remobilized layer in the near bed, b) the fastest front velocities (up to 8.1 m/s) ever measured from instruments yet; c) the fact that major triggering mechanisms such as earthquakes, river floods or large wave heights are not necessary conditions for the triggering of turbidity currents, rather that small wave heights may cause seafloor failure that generate turbidity currents.

The topic is of great practical and scientific interest for a wide audience, as it is relevant in various context spanning from interpreting the architecture of ancient sedimentary deposits (geology), threatening the safety of submarine infrastructures (engineering) or understanding the effects of such flows on the disturbance of biological communities (ecology), and many others. This notwithstanding, only a few direct flow and concentration measurements are available in the literature as turbidity currents are extremely challenging to be measured.

The objectives of the paper are clearly explained, the results are properly presented and the discussion of the results is clear and quite convincing. Along with this mostly favourable reaction, I do have some minor reservations concerning further analyses that would help strengthen the conclusions of the paper. Namely:

1. I acknowledge that the monitoring study is extremely ambitious, as an unprecedented number of sensor (>50) is deployed on the seafloor for a long period of time (18 months). This notwithstanding, one of the major claim of the paper, namely the presence of the dense remobilized layer, is inferred from the repeated movement of a buried 2m tall frame rather than directly detected with any of the sensors employed. Drag on the foot of the tripod frame and rotation of the instrument might lead to exposure and transport downstream. In this respect, I agree with the authors that the motion of the instruments and their anchors is better explained by the anchors being rafted in a dense layer rather than by drag. When I first read the paper I was hoping to get to a point where the velocity profile of the turbidity current was measured in the near-bed region or even below the seafloor to capture the sand motion within the remobilized layer. With this consideration, I do not want to limit the success of the observations. Rather I am asking if there are not other evidences available to further demonstrate that during flows the dense remobilized layer is indeed present. For example, I was wondering if there are not evidences that any BED instrument was first buried and then later remobilized. If available that observation would further reinforce the dense layer hypothesis, not conclusively though, as erosion and deposition during the turbidity current event might lead to exposure, transport and burial of the instrument as

well.

2. Figure 3 is crucial to demonstrate that between maximum wave heights and occurrence of turbidity currents are poorly correlated. The quality of the plot is not excellent, and in view of the relevance of the figure I would suggest to improve it.
3. In order to further strengthen the fact that turbidity currents are not necessarily associated with external forcing, it would be great to add a figure showing that the turbidity current events are not correlated with floods. The authors might refer to either rainfall data in the most proximal river basin or to river flow discharge data if available.
4. Some additional details (maybe as a supplementary material) should be included for the reader to have a more complete view of the observations. I would recommend including: i) repeated surveys and difference maps of the seabed in order to show the erosional/depositional pattern along the canyon, the direction of migration of bedforms if any along the canyon, channel widening/narrowing, etc... ii) one or more tables with a synthesis of the various observed flows, their runoff, duration of the event, velocity measured by ADCPs and BEDS, wave height, approximate flow thickness, suspended sediment concentration, etc...

Editorial Comments

line 30: "m/sec" → "m/s".

line 156: "(Figure 4D" → "(Figure 4D)".

line 165: " ρ _{in water}" is not defined.

line 402: Remove reference Shanmiugam G. as it is already present (line 385).

Figure 1: Add vertical scale to give an idea of the height above the seabed of the instruments.

Figure 5: All instruments were transported downstream with the only exception of BED03 that moved upstream. Correct? If that is the case, what is the explanation for that? Some comment is needed in the text to discuss this unexpected behaviour.

Michele Bolla Pittaluga

Reviewer #2 (Remarks to the Author):

General assessment:

I reviewed the manuscript entitled “Anatomy of powerful turbidity currents revealed by most detailed monitoring yet.” The authors claim that oceanic turbidity currents may be generated by liquefaction to form a dense basal layer beneath a dilute turbidity cloud. The authors proposed a new model that challenges the current theory that usually refers the flows are mostly triggered by external events (storms, floods or earthquake). The authors also claim that they used the most detailed monitoring data to support their model/hypothesis/observations.

I generally agree with their points. The data is unique and supportive. The existence of a dense basal layer beneath a dilute turbidity current may play an important role for shaping the floor of submarine canyons. Although the velocity and concentration data are not directly measured from ADCP, the authors used Benthic Event Detectors (BEDs) and Autonomous Underwater Vehicles (AUVs) to overcome this limitation.

For field measurement of oceanic turbidity currents, their claim are novel. Actually, some of the small-scaled laboratory experiments do support their points, e.g. Lai et al. (2016) in GRL, the authors have shown that the thin, dense saline currents have the ability to trigger erosion processes and generate bedload transport. In their experiments, Lai et al. (2016) do report the observations of flowsliding/shallow dense sliding happening as a part of submarine canyon evolution, from canyon walls and along a canyon thalweg. These processes are actually similar to the ideas of “dense basal layer.” I believe in laboratory- or field-scale the dense basal layer may play an important role contributing to the change of bed morphology, not dilute turbidity currents.

However, how to generate the basal dense layer? Currently, I am still not convinced by the liquefaction mechanism proposed by the authors. In other words, along shore currents (Lastras et al., 2011), floods, storms, earthquake or dense shelf cascading flows (Canals et al., 2006) may also generate a dense basal layer beneath the sediment gravity current. The current implications for liquefaction-induced dense basal layers are too strong/far, unless the authors can provide direct evidence/measurement of liquefaction and its related sediment transport processes.

I believe this paper is important and of interest to ocean science. Especially, the proposed model of dense basal layer may provide an alternative way to look at the profile of turbidity currents. Overall, the manuscript is clearly written. Most of the figures are well presented. I enjoy reading the manuscript. My specific comments for the text and figures are as list below, I hope these comments may help the authors improve their current manuscript. I encourage the authors resubmit their manuscript after a minor revision.

Specific comments:

[Line 156] Typo, it needs a bracket.

[Line 215] The implications of liquefaction is too strong. I suggest the authors either provide directly evidence or rephrase the paragraph.

[Figure 1]

Figure 1 is a very good demonstration for the installation of these measuring devices, especially for non-oceanic researchers.

[Figure 2]

The presentation of Figure 2 needs a large improvement. The data deserves a better presentation. I suggest using a similar style of Figures 6, 7, and 9 of Clarke, J. E. H. (2016), Nat Common. It may help raising the clarity and quality of the current Figure 2. My specific suggestions/questions are list below.

(1) What do the white values/numbers in Figure 2a mean? What are their units?

(2) The same, what do the black values/numbers in Figure 2a mean?

(3) The symbol of measuring device (e.g. locations of MS1, MS2...etc.) are too small to see in the map of Monterey Canyon. I suggest not using dash lines to indicate their locations. It is distracting and not easy to see when the dash lines are too thin.

(4) Place a color bar on each ADCP and backscatter sub-figures (as the Figure 6 of Clarke, J. E. H., 2016); or alternatively place two individual colorbars for ADCP and backscatter, respectively (as the Figure 9 of Clarke, J. E. H., 2016). The range of the colorbars should be consistent; otherwise, it is easy to misleading the readers.

(5) For the layout of Figure 2, would it be possible to redesign it by using the following alternative style?

(a) Three vertical columns, place the map on the left (Figure 2a), ADCP velocity in the middle (Figure 2b) and backscatters on the right (Figure 2c). Therefore, the map can be enlarged to earn more space to place clear symbols for those measurement devices.

(b) Place the map horizontally in the upper panel (i.e. the canyon head is on the right and the deep-sea plane is on the left) (Figure 2a); place the ADCP velocity on the lower left and the backscatter on the lower right panel.

[Figure 3]

I believe the authors tried to use Figure 3 to tell a time story of turbidity currents. I suggest shorten the height of Figure 3a. Save more space for the data of maximum wave heights (Figure 3b).

Currently, the axes of Figure 3b are too thin, not consistent with Figure 3a. The dates of its x-axis are too close, I suggest resize the letters and add minor ticks on the x-axis. I suggest place the same x-axis (with minor ticks, but without dates) in the bottom of Figure 3a. Therefore, the readers can easily follow the time events. I cannot see the "thin red lines" in Figure 3b as mentioned in the legend. For the gray vertical lines in Figure 3b, I suggest changing the color to a brighter, contrasting color (e.g. pink, light green...etc. or simply red).

[Figure 4]

Currently, Figure 4 is too busy and the quality is not yet to the journal's level. The authors try to put too many information in the same figure. I suggest cut Figure 4b to earn more space for the measured data. If the authors are reluctant to cut the images/cartoons of Figure 4b. I suggest insert these images in Figure 1 somewhere, because they are similar. The authors can easily refer it to the Figure. The colorbars of Figure 4a are not aligned. The data in Figures 4c and 4d are too busy/crowded. If they agree to cut Figure 4b, the authors may place the these figures in a vertical column (upper, middle and lower panel) to make Figures 4c and 4d a bit wider, therefore the readers may have chance to see clearly how the data are performed, especially in the blue box of Figure 4d. Does the direction of the gray arrows in Figure 4c mean something? The style of the dates placed in Figure 4d should be consistent with others.

[Figure 5]

The moving paths in Figure 5b are too thin, almost impossible to see. I suggest using thicker color solid lines. Similar problem, the authors used thin dash lines to mark the locations of the sub-figures x, y, z. I suggest simply using thicker black solid boxes.

[Figure 6]

I suggest directly labeling Stages 1, 2, 3 on Figure 6. Currently, I do not really understand what the authors are referring. Finally, for a better follow-up of their model, the authors may color the box in Figure 6c-1 and Figure 6b, in red and blue, respectively. Therefore, the readers may easily feel the different scales.

Steven Yueh Jen Lai

Reviewer #3 (Remarks to the Author):

Dear Editors,

The paper “Anatomy of powerful turbidity currents revealed by most detailed monitoring yet” by Paull et al is an excellent, data-rich contribution that documents multiple sediment transport events in a submarine canyon and provides critical information about the structure and rheology of this flow. I think this contribution is worthy of publication in Nature Communications with minor revisions. I detail the main issues that need to be addressed, and the pdf has many in-line comments. Please see the attached word document with better formatting than this webform.

Thanks,

Zane Jobe

Colorado School of Mines

Main issues to address:

1. This paper describes a ‘dense remobilized layer’ and proposes this layer as a major mode of sediment transport in turbidity currents. This can be corrected by moving arguments from the supplement into the main document and better relating the authors’ observations to recent field and numerical modeling efforts.
 - a. Even with the revolutionary data collected, no direct measurements are made in this layer and many uncertainties remain; so, some of the statements need to be softened/caveated. Also, the paper does not include recent, key work on numerical models of this layer (Eggenhuisen et al 2017, Vellinga et al 2018) as well as other papers proposing near-bed high sediment concentration (e.g., Hansen et al 2015, Jobe et al 2017).
 - b. Importantly, linkages are made between this layer and the bedforms found all around the world in submarine canyons, but are only made in the supplementary data. These arguments need to be brought into the main manuscript and expanded, citing (brand) new work (e.g., Hage et al 2018). When proposing a new model, the main argument cannot be in the supplement. I also recommend a short discussion of sub- and super-critical flows, as these bedforms are commonly linked to hydraulic jumps in the flow. There is a burgeoning literature on these flows and associated bedforms, and this paper needs to place itself within that literature. I have commented where

appropriate to insert references to recent papers (e.g., Cartigny et al. 2014, Postma et al. 2014, Symons et al., 2016, Hage et al. 2018, Vellinga et al 2018).

2. The argument that no trigger is necessary to cause a flow-event in the canyon is kind of a straw-man argument. This can be easily corrected by softening the argument and citing relevant papers.

a. It is true that this paper documents 1 or 2 flows that have no apparent wave-trigger, but this is not a new idea. There are at least a few papers that have documented flow events happening with no apparent trigger (e.g., Carter et al 2012, maybe Paull et al 2010?). Both of these papers are authored/coauthored by these same authors of this manuscript, and I am surprised to not see them cited here.

b. When I say straw-man, I mean that the authors build up an argument that the community thinks that flows always have a recognizable trigger (e.g., earthquake, wave action, flood, etc.). They then knock that straw-man down by saying that they document flows when no trigger happens. But, as stated above, there is ample evidence of flows with no trigger in the literature.

c. Also, the authors do not investigate other possible triggers (e.g., minor earthquake, river flood). It would be nice to see this in the paper to strengthen their argument of no trigger.

d. Lastly, and perhaps most important - the vast majority of the documented events DO occur with a clear trigger (high wave activity), indicating a strong correlation between a flow event and a trigger. I am not trying to say that the authors are wrong, just that they are over-emphasizing the argument of flows with no trigger, when 90% of their data shows that triggers do cause flows.

3. As noted above, the paper needs to cite recent, relevant references throughout the manuscript.

4. Generally the figures are good, but I have made numerous small in-line comments to improve the figures for a general (non-expert) audience. Making these figures as clear and easy to digest as possible will be important to help this paper make the splash it deserves to make.

a. Fig 2 – different time reference between ‘a’ and ‘b’; labeling and color bar

b. Fig 4 – minor tweaks and label additions to clarify arguments

c. Fig 5 and 6 – labeling

References:

Cartigny et al 2014 <https://doi.org/10.1111/sed.12076>

Carter et al 2012 <https://doi.org/10.1029/2012GL051172>

Eggenhuisen et al 2017 doi:10.5194/esurf-5-269-2017

Hage et al 2018 <https://doi.org/10.1130/G40095.1>

Hansen et al 2015 <https://doi.org/10.1016/j.marpetgeo.2015.06.007>

Jobe et al 2017 <https://doi.org/10.1002/2016JF003903>

Paull et al 2010 <https://doi.org/10.1130/GES00527.1>

Postma et al 2014 <https://doi.org/10.1111/sed.12135>

Symons et al 2016 <https://doi.org/10.1016/j.margeo.2015.11.009>

Vellinga et al 2018 <https://doi.org/10.1111/sed.12391>

REVIEWERS' COMMENTS:

Reviewer #1 (Remarks to the Author): Overall Assessment

The paper reports the first results from an ambitious monitoring study conducted in Monterey Canyon (offshore California) revealing new insights on the behaviour of turbidity currents. The major claims of the paper include: a) the attempt to unravel the presence of a dense, fast, remobilized layer in the near bed, b) the fastest front velocities (up to 8.1 m/s) ever measured from instruments yet; c) the fact that major triggering mechanisms such as earthquakes, river floods or large wave heights are not necessary conditions for the triggering of turbidity currents, rather that small wave heights may cause seafloor failure that generate turbidity currents.

The topic is of great practical and scientific interest for a wide audience, as it is relevant in various context spanning from interpreting the architecture of ancient sedimentary deposits (geology), threatening the safety of submarine infrastructures (engineering) or understanding the effects of such flows on the disturbance of biological communities (ecology), and many others. This notwithstanding, only a few direct flow and concentration measurements are available in the literature as turbidity currents are extremely challenging to be measured.

The objectives of the paper are clearly explained, the results are properly presented and the discussion of the results is clear and quite convincing. Along with this mostly favourable reaction, I do have some minor reservations concerning further analyses that would help strengthen the conclusions of the paper. Namely:

1. I acknowledge that the monitoring study is extremely ambitious, as an unprecedented number of sensor (>50) is deployed on the seafloor for a long period of time (18 months). This notwithstanding, one of the major claim of the paper, namely the presence of the dense remobilized layer, is inferred from the repeated movement of a buried 2m tall frame rather than directly detected with any of the sensors employed. Drag on the foot of the tripod frame and rotation of the instrument might lead to exposure and transport downstream. In this respect, I agree with the authors that the motion of the instruments and their anchors is better explained by the anchors being rafted in a dense layer rather than by drag. When I first read the paper I was hoping to get to a point where the velocity profile of the turbidity current was measured in the nearbed region or even below the seafloor to capture the sand motion within the remobilized layer. With this consideration, I do not want to limit the success of the observations. Rather I am asking if there are not other evidences available to further demonstrate that during flows the dense remobilized layer is indeed present. For example, I was wondering if there are not evidences that any BED instrument was first buried and then later remobilized. If available that observation would further reinforce the dense layer hypothesis, not conclusively though, as erosion and deposition during the turbidity current event might lead to exposure, transport and burial of the instrument as well.

We now clearly state on line 234 that sediment concentration was not measured directly, and that it has to be inferred using other evidence (i.e., line 115).

The reviewer asked if there was further evidence that the BEDS were buried and then re-exhumed. To address this, we have added that this behaviour has also been seen previously with prototypes of the BEDs. Paull et al., 2010 showed that similar sized objects (i.e., monuments) which were predecessors to the BEDs were shown to be buried by ≥ 1 m of sediment and then move similar distances down the canyon multiple times during a 2-year time period. We have also added to the text that objects with very different sizes, densities and shapes moved at broadly similar speeds down-canyon. This observation is more consistent with rafting than being dragged beneath a dilute flow, where such objects would be expected to travel at different speeds.

2. Figure 3 is crucial to demonstrate that between maximum wave heights and occurrence of turbidity currents are poorly correlated. The quality of the plot is not excellent, and in view of the relevance of the figure I would suggest to improve it.

See comment 3 below.

3. In order to further strengthen the fact that turbidity currents are not necessarily associated with

external forcing, it would be great to add a figure showing that the turbidity current events are not correlated with floods. The authors might refer to either rainfall data in the most proximal river basin or to river flow discharge data if available.

Comments 2 & 3 are directed to Figure 3. Which has been modified to improve the clarity and to include the discharge of the Salinas River, which enters the ocean directly at the head of the Monterey Canyon.

4. Some additional details (maybe as a supplementary material) should be included for the reader to have a more complete view of the observations. I would recommend including: i) repeated surveys and difference maps of the seabed in order to show the erosional/depositional pattern along the canyon, the direction of migration of bedforms if any along the canyon, channel widening/narrowing, etc... ii) one or more tables with a synthesis of the various observed flows, their runout, duration of the event, velocity measured by ADCPs and BEDS, wave height, approximate flow thickness, suspended sediment concentration, etc...

Supplementary materials have been added which contain some of these materials, including data files pertaining to the timing of events. Figure 2a, and larger-format Figure 5 do show, the canyon morphology (including widening) and morphological changes. In some cases, we do not have provided data to address these points. For example, our repeat bathymetry surveys are too widely spaced in time and the changes between surveys too large to confidently determine migration direction. Moreover, several of these requests are neither relevant to the themes addressed in this paper, nor possible within the scope of the methods presented in this paper.

Editorial Comments

line 30: "m/sec" → "m/s".

line 156: "(Figure 4D" → "(Figure 4D)".

line 165: "p _{in water}" is not defined.

line 402: Remove reference Shanmiugam G. as it is already present (line 385).

All these editorial comments were accommodated (although some became irrelevant through other modifications).

Figure 1: Add vertical scale to give an idea of the height above the seabed of the instruments.

Figure 1. The drawing is not to scale. However, to accommodate this we have added that "Depth above seafloor for key instruments are indicated" to the caption, and inserted the number in the figure.

Figure 5: All instruments were transported downstream with the only exception of BED03 that moved upstream. Correct? If that is the case, what is the explanation for that? Some comment is needed in the text to discuss this unexpected behaviour.

Figure 5. The reviewer was confused by an arrow oriented upwards. This was intended to indicate that BED 3 floated to the surface. To address this, we removed the arrow and added 'dropped ballast' to the BED 3 label.

Michele Bolla Pittaluga

Reviewer #2 (Remarks to the Author):

General assessment:

I reviewed the manuscript entitled "Anatomy of powerful turbidity currents revealed by most detailed monitoring yet." The authors claim that oceanic turbidity currents may be generated by liquefaction to form a dense basal layer beneath a dilute turbidity cloud. The authors proposed a new model that challenges the current theory that usually refers the flows are mostly triggered by external events

(storms, floods or earthquake). The authors also claim that they used the most detailed monitoring data to support their model/hypothesis/observations.

I generally agree with their points. The data is unique and supportive. The existence of a dense basal layer beneath a dilute turbidity current may play an important role for shaping the floor of submarine canyons. Although the velocity and concentration data are not directly measured from ADCP, the authors used Benthic Event Detectors (BEDs) and Autonomous Underwater Vehicles (AUVs) to overcome this limitation.

For field measurement of oceanic turbidity currents, their claim are novel. Actually, some of the small-scaled laboratory experiments do support their points, e.g. Lai et al. (2016) in GRL, the authors have shown that the thin, dense saline currents have the ability to trigger erosion processes and generate bedload transport. In their experiments, Lai et al. (2016) do report the observations of flowsliding/shallow dense sliding happening as a part of submarine canyon evolution, from canyon walls and along a canyon thalweg. These processes are actually similar to the ideas of "dense basal layer." I believe in laboratory or fieldscale the dense basal layer may play an important role contributing to the change of bed morphology, not dilute turbidity currents.

We have added a reference to the paper in the Introduction, where other modeling and architecture issues are considered.

However, how to generate the basal dense layer? Currently, I am still not convinced by the liquefaction mechanism proposed by the authors. In other words, along shore currents (Lastras et al., 2011), floods, storms, earthquake or dense shelf cascading flows (Canals et al., 2006) may also generate a dense basal layer beneath the sediment gravity current. The current implications for liquefaction-induced dense basal layers are too strong/far, unless the authors can provide direct evidence/measurement of liquefaction and its related sediment transport processes.

We agree that once generated the events discussed in Lastras et al., 2011 and Canals et al., 2006 may have been associated with a dense basal layer. However, these (excellent) past papers provide few, if any, constraints on whether dense layers were present in their flows, so we just do not know. Importantly, these previous studies did not provide evidence for dense layers, such as is done here using tracking of heavy objects with embedded sensors. They lacked the type of near-bed observations included here.

This is a reasonable point. In the rewritten text and modification to Figure 3 we have softened the discussion of liquefaction, and acknowledge uncertainties. For example, we say that (line 289): "*Suitable in-situ physical properties measurements (e.g. pore pressure^{44,45}) to determine exact processes of erosion and bedform generation near the flow front were not collected.*" We also now cite recent seminal large-scale experiments and theoretical work by Iverson and others, which support a view that liquefaction would be likely in these full-saturated sands.

I believe this paper is important and of interest to ocean science. Especially, the proposed model of dense basal layer may provide an alternative way to look at the profile of turbidity currents. Overall, the manuscript is clearly written. Most of the figures are well presented. I enjoy reading the manuscript. My specific comments for the text and figures are as list below, I hope these comments may help the authors improve their current manuscript. I encourage the authors resubmit their manuscript after a minor revision.

Specific comments:

[Line 156] Typo, it needs a bracket.

[Line 215] The implications of liquefaction is too strong. I suggest the authors either provide directly evidence or rephrase the paragraph.

All the specific editorial corrections were addressed. However, the reformatting made them obsolete.

[Figure 1]

Figure 1 is a very good demonstration for the installation of these measuring devices, especially for nonoceanic researchers.

[Figure 2]

The presentation of Figure 2 needs a large improvement. The data deserves a better presentation. I suggest using a similar style of Figures 6, 7, and 9 of Clarke, J. E. H. (2016), Nat Common. It may help raising the clarity and quality of the current Figure 2. My specific suggestions/questions are list below.

(1) What do the white values/numbers in Figure 2a mean? What are their units?

See # 2 below

(2) The same, what do the black values/numbers in Figure 2a mean?

(Points 1 & 2). Apparently, the meaning of the white and black numbers in Figure 2A did not come through. While these numbers were explained in the caption, we have modified the caption to make this clearer, including adding a label for the Transit Velocity column.

(3) The symbol of measuring device (e.g. locations of MS1, MS2...etc.) are too small to see in the map of Monterey Canyon. I suggest not using dash lines to indicate their locations. It is distracting and not easy to see when the dash lines are too thin.

The size of the symbols for the moorings in the canyon axis have been increased. The lines connecting between the moorings on the map and the ADCP data have been removed and replaced with mooring labels (MS1 to MS7, SIN). We have also labeled Soquel Canyon, as it is now mentioned in the text.

(4) Place a color bar on each ADCP and backscatter subfigures (as the Figure 6 of Clarke, J. E. H., 2016); or alternatively place two individual colorbars for ADCP and backscatter, respectively (as the Figure 9 of Clarke, J. E. H., 2016). The range of the colorbars should be consistent; otherwise, it is easy to misleading the readers.

If we show the ADCP velocity for the two events at the same scale, we will not see the smaller September 1 event, hopefully the scalebars are more noticeably different now.

The backscatter pattern has been changed from multicolored to black and white.

(5) For the layout of Figure 2, would it be possible to redesign it by using the following alternative style? (a) Three vertical columns, place the map on the left (Figure 2a), ADCP velocity in the middle (Figure 2b) and backscatters on the right (Figure 2c). Therefore, the map can be enlarged to earn more space to place clear symbols for those measurement devices.

(b) Place the map horizontally in the upper panel (i.e. the canyon head is on the right and the deepsea plane is on the left) (Figure 2a); place the ADCP velocity on the lower left and the backscatter on the lower right panel.

Without understanding what the meaning of the numbers in this plot, the main point of the figure did not come through to this reviewer. While we considered following this suggestion, we feel that it would not be an improvement. In fact, the most important feature, that the transit velocities are usually higher than the highest reliable instantaneous measurement made by the ADCPs, would not come through.

[Figure 3]

I believe the authors tried to use Figure 3 to tell a time story of turbidity currents. I suggest shorten the height of Figure 3a. Save more space for the data of maximum wave heights (Figure 3b).

Currently, the axes of Figure 3b are too thin, not consistent with Figure 3a. The dates of its xaxis are too close, I suggest resize the letters and add minor ticks on the xaxis. I suggest place the same xaxis (with minor ticks, but without dates) in the bottom of Figure 3a. Therefore, the readers can easily follow the time events. I cannot see the "thin red lines" in Figure 3b as mentioned in the legend. For

the gray vertical lines in Figure 3b, I suggest changing the color to a brighter, contrasting color (e.g. pink, light green...etc. or simply red).

Figure 3 – We have replaced Figure 3 with a new version, partly along these lines suggested here and partly along the lines suggested by Reviewer 3.

[Figure 4]

Currently, Figure 4 is too busy and the quality is not yet to the journal's level. The authors try to put too many information in the same figure. I suggest cut Figure 4b to earn more space for the measured data. If the authors are reluctant to cut the images/cartoons of Figure 4b. I suggest insert these images in Figure 1 somewhere, because they are similar. The authors can easily refer it to the Figure. The colorbars of Figure 4a are not aligned. The data in Figures 4c and 4d are too busy/crowded. If they agree to cut Figure 4b, the authors may place the these figures in a vertical column (upper, middle and lower panel) to make Figures 4c and 4d a bit wider, therefore the readers may have chance to see clearly how the data are performed, especially in the blue box of Figure 4d. Does the direction of the gray arrows in Figure 4c mean something? The style of the dates placed in Figure 4d should be consistent with others.

To make this less busy, we have opened up the figure and remove the image of the square BED. The time scale on parts a and c are the same, and the depth scale on parts c and d are the same. The grey highlights in part a and c indicate the interval during which all six BEDS were moving simultaneously. The pink band highlights the interval in c and d when the AMT frame carrying BED11 moved down canyon.

[Figure 5]

The moving paths in Figure 5b are too thin, almost impossible to see. I suggest using thicker color solid lines. Similar problem, the authors used thin dash lines to mark the locations of the subfigures x, y, z. I suggest simply using thicker black solid boxes.

The lines indicating pathway of equipment moving on the canyon floor and red boxes have been thickened.

[Figure 6]

I suggest directly labeling Stages 1, 2, 3 on Figure 6. Currently, I do not really understand what the authors are referring. Finally, for a better followup of their model, the authors may color the box in Figure 6c1 and Figure 6b, in red and blue, respectively. Therefore, the readers may easily feel the different scales.

Labels for 'Stages' 1-4 have been added as suggested. Red and blued colors for the boxes in parts 6b and c Stage 1 have been added as suggested.

Steven Yueh Jen Lai

Reviewer #3 (Remarks to the Author):

Dear Editors,

The paper "Anatomy of powerful turbidity currents revealed by most detailed monitoring yet" by Paull et al is an excellent, datarich contribution that documents multiple sediment transport events in a submarine canyon and provides critical information about the structure and rheology of this flow. I think this contribution is worthy of publication in Nature Communications with minor revisions. I detail the main issues that need to be addressed, and the pdf has many inline comments. Please see the attached word document with better formatting than this webform.

Thanks,

Zane Jobe

Colorado School of Mines

Main issues to address:

1. This paper describes a 'dense remobilized layer' and proposes this layer as a major mode of

sediment transport in turbidity currents. This can be corrected by moving arguments from the supplement into the main document and better relating the authors' observations to recent field and numerical modeling efforts.

We agree, and have moved the essence of what was in supplemental materials into the text. (The original manuscript was submitted to Nature Geoscience, which has a shorter format, and then redirected by Nature Geoscience to Nature Communications).

a. Even with the revolutionary data collected, no direct measurements are made in this layer and many uncertainties remain; so, some of the statements need to be softened/caveated.

We have also tried to soften the wording on the certainty about the dense layers. For example, on Line 224 we make it clear that we lack direct concentration measurements, and those concentrations need to be inferred from a variety of other data types.

Also, the paper does not include recent, key work on numerical models of this layer (Eggenhuisen et al 2017, Vellinga et al 2018) as well as other papers proposing nearbed high sediment concentration (e.g., Hansen et al 2015, Jobe et al 2017).

These references have been added.

b. Importantly, linkages are made between this layer and the bedforms found all around the world in submarine canyons, but are only made in the supplementary data. These arguments need to be brought into the main manuscript and expanded, citing (brand) new work (e.g., Hage et al 2018). When proposing a new model, the main argument cannot be in the supplement. I also recommend a short discussion of sub and supercritical flows, as these bedforms are commonly linked to hydraulic jumps in the flow. There is a burgeoning literature on these flows and associated bedforms, and this paper needs to place itself within that literature. I have commented where appropriate to insert references to recent papers (e.g., Cartigny et al. 2014, Postma et al. 2014, Symons et al., 2016, Hage et al. 2018, Vellinga et al 2018).

We have added these references to more explicitly addresses the sub and supercritical flow topic. We also moved the discussion about bedforms into the main text and added a figure relevant to this.

2. The argument that no trigger is necessary to cause a flowevent in the canyon is kind of a strawman argument. This can be easily corrected by softening the argument and citing relevant papers.

a. It is true that this paper documents 1 or 2 flows that have no apparent wavetrigger, but this is not a new idea. There are at least a few papers that have documented flow events happening with no apparent trigger (e.g., Carter et al 2012, maybe Paull et al 2010?). Both of these papers are authored/coauthored by these same authors of this manuscript, and I am surprised to not see them cited here.

As the reviewer notes, we were involved in the Carter et al., and Paull et al., papers. The Carter et al. 2012 paper actually demonstrates that turbidity currents were associated with a river flood, but that there was a delay of ~3 days, most likely as flood sediment accumulated and then failed. Thus, the reviewer is incorrect here, this turbidity current was linked to a flood, albeit with a delay. Paull et al. 2010 also link flow events to storms, albeit in a more general way. But we now note the potential for delays of up to several days after a major perturbation such as a river flood (lines 86-87).

b. When I say strawman, I mean that the authors build up an argument that the community thinks that flows always have a recognizable trigger (e.g., earthquake, wave action, flood, etc.). They then knock that strawman down by saying that they document flows when no trigger happens. But, as stated above, there is ample evidence of flows with no trigger in the literature.

Please see our answer above, where the reviewer cites the Carter et al. 2012 and Paull et al. 2010 papers. Those papers do not say that turbidity currents occur when no major external

trigger happens. For example, in Carter et al. 2012, the turbidity current is delayed but still associated with a river flood. We have written reviews on this topic [Talling, Paull, Piper, 2014, ESR], but know of no other examples of papers that say there is no trigger.

c. Also, the authors do not investigate other possible triggers (e.g., minor earthquake, river flood). It would be nice to see this in the paper to strengthen their argument of no trigger.

This is a good point, and we now include a plot of river discharge to back up our conclusions.

d. Lastly, and perhaps most important the vast majority of the documented events DO occur with a clear trigger (high wave activity), indicating a strong correlation between a flow event and a trigger. I am not trying to say that the authors are wrong, just that they are over emphasizing the argument of flows with no trigger, when 90% of their data shows that triggers do cause flows.

This is a very good point by the reviewer. We now make it clear that the majority of events do appear to have a major external trigger (see line 219-221 – *“Fourteen of the fifteen flows occurred in the winter months (Fig. 3). These events typically coincide with large storm wave heights (Fig. 3b), which may have triggered seabed failure in the upper canyon.”*).

3. As noted above, the paper needs to cite recent, relevant references throughout the manuscript.

We have added many recent references.

4. Generally the figures are good, but I have made numerous small inline comments to improve the figures for a general (nonexpert) audience. Making these figures as clear and easy to digest as possible will be important to help this paper make the splash it deserves to make.

a. Fig 2 – different time reference between ‘a’ and ‘b’; labeling and color bar

Again, this a perceptive comment. The velocity and backscatter plots intentionally are on different time lines. In part a they are the same time window for both the January 15th and September 1st event. This is to emphasis the passage of the event down canyon. However, in part b, the goal is to show something about how the flow evolves downcanyon. These start just before the arrival of the flows. By changing the backscatter data in part b to a black and white color scale, the confusion about the scale bars and the labeling is corrected. The lines connecting between the moorings on the map and the ADCP data have been removed and replaced with mooring labels (MS1 to MS7, SIN).

b. Fig 4 – minor tweaks and label additions to clarify arguments

We have changed the color scale for the backscatter to black and white, both here and in figure 2. We added arrows pointing to the AMT foot, as requested in marked -up pdf. We have also extended the grey area upward in C to connect with same time period in A. Similarly, we have indicated that the depth range of interest in part C and D are the same using a pink area.

c. Fig 5 and 6 – labeling

While there was a label for CSB in the figure 5, it was small. Thus, we have enlarged its text size to make it easier to see. Also, have added the dates of the AUV mapping surveys to Figure 5 as requested. We have added ‘Stage’ labels to Figure 5.